# The Effect of Travel Burden on Depression and Anxiety in African American Women Living with Systemic Lupus

**DOI:** 10.3390/healthcare9111507

**Published:** 2021-11-05

**Authors:** Ashley A. White, Brittany L. Smalls, Aissatou Ba, Trevor D. Faith, Viswanathan Ramakrishnan, Hetlena Johnson, Jillian Rose, Clara L. Dismuke-Greer, Jim C. Oates, Leonard E. Egede, Edith M. Williams

**Affiliations:** 1Department of Public Health Sciences, Medical University of South Carolina, 135 Cannon Street, Suite 301, Charleston, SC 29425, USA; whitashl@musc.edu (A.A.W.); baaai@musc.edu (A.B.); ramakris@musc.edu (V.R.); 2Department of Family and Community Medicine, University of Kentucky College of Medicine, 2195 Harrodsburg Road, Suite 125, Lexington, KY 40504, USA; brittany.smalls@uky.edu; 3Biomedical Informatics Center, Medial University of South Carolina, 135 Cannon Street, Suite 101, Charleston, SC 29425, USA; faitht@musc.edu; 4Lupus CSC, Columbia, SC 29210, USA; hjohnson@lupuscsc.org; 5Community Engagement, Diversity & Research, Department of Social Work Programs, Hospital for Special Surgery, 535 East 70th Street, New York, NY 10021, USA; rosej@hss.edu; 6Health Economics Resource Center, Veterans Administration Palo Alto Health System, 795 Willow Road, Menlo Park, CA 94025, USA; clara.dismuke@va.gov; 7Department of Medicine, Division of Rheumatology, Medical University of South Carolina, 96 Jonathan Lucas Street, Charleston, SC 29425, USA; oatesjc@musc.edu; 8Division of General Internal Medicine, Center for Patient Care and Outcomes Research, Medical College of Wisconsin, Milwaukee, WI 53226, USA; legede@mcw.edu

**Keywords:** systemic lupus erythematosus (SLE), PHQ-8, GAD-8, anxiety, depression

## Abstract

The United States has a deficit of rheumatology specialists. This leads to an increased burden in accessing care for patients requiring specialized care. Given that most rheumatologists are located in urban centers at large hospitals, many lupus patients must travel long distances for routine appointments. The present work aims to determine whether travel burden is associated with increased levels of depression and anxiety among these patients. Data for this study were collected from baseline visits of patients participating in a lupus study at MUSC. A travel/economic burden survey was assessed as well as the 8-item Patient Health Questionnaire (PHQ-8) and the 7-item Generalized Anxiety Disorder (GAD-7) survey as measures of depression and anxiety, respectively. Linear regression models were used to assess the relationship between travel burden and depression and anxiety. Frequency of healthcare visits was significantly associated with increased depression (β = 1.3, *p* = 0.02). Significant relationships were identified between anxiety and requiring time off from work for healthcare appointments (β = 4, *p* = 0.02), and anxiety and perceived difficulty in traveling to primary care providers (β = 3.1, *p* = 0.04). Results from this study provide evidence that travel burden can have an effect on lupus patients’ anxiety and depression levels.

## 1. Introduction

It is estimated that 1.5 million Americans have some form of lupus [1]. Globally, systemic lupus erythematosus (SLE) is estimated to effect at least five million individuals, with more than 100,000 new cases developing every year [2,3,4,5]. SLE and other autoimmune diseases are identified when one’s immune system loses its ability to differentiate between foreign substances and its own cells and tissues, causing the body to attack itself. SLE has a wide spectrum of clinical presentations, which are characterized by remissions and exacerbations. SLE can affect any part of the body (e.g., skin, joints, blood, and kidneys) and can be life-threatening [6]. Vital organs are affected by lupus, including alterations in the heart, lungs, and brain, which can result in morbidity for patients. In the United States, the highest lupus morbidity and mortality rates are among African American women [7,8,9]. Specifically, SLE affects approximately 1 in 250 African American women of childbearing age, who have three to four times higher prevalence of SLE; are at risk for developing SLE at an earlier age; and have increased SLE-related disease activity, damage, and mortality compared to White Americans [10,11,12,13].

Unfortunately, healthcare providers who are well-versed in the care for those diagnosed with SLE are sparse (e.g., rheumatologists). In an assessment of rheumatologists’ workforce in 2015, there were 4497 board certified rheumatologists [14]. However, there was an estimated shortage of 1118 rheumatologists based on patient need for this specialty [14]. Moreover, it is projected that by 2030, there will be a demand for an estimated 8149 rheumatologists [14]. Even with an increase in board certified rheumatologists, there is a projected shortage of rheumatologist of 4729 [14]. The shortage of rheumatologists makes health-related travel increasingly difficult for SLE patients who require their care.

In previous research, health-related travel has been investigated as a potential barrier to accessing health care and has been shown to lead to rescheduled or missed appointments, delayed care, and low medication adherence [15,16]. Low medication adherence, particularly among African American women, has been shown to be associated with depressive symptoms [17]. Depression has been shown to have a significant positive correlation with organ damage in African American women with SLE [18]. Furthermore, compared to White Americans, African Americans are more likely to experience psychosocial stressors, and depression can increase levels of stress and anxiety, which then may induce flares and worsen SLE symptoms [19,20]. However, there is limited information on whether travel burden to receive health care influences depression and anxiety among those diagnosed with SLE [15,20]. Therefore, the objective of this study was to evaluate the impact of travel burden on depression and anxiety among African American women with SLE.

## 2. Materials and Methods

For the current study, data from a study that aimed to test the effectiveness of socio-behavioral interventions for SLE patients in South Carolina were utilized. Only baseline (pre-intervention) data were utilized for this study to negate the potentially confounding impacts the interventions may have had on the treatment groups. The Peer Approaches to Lupus Self-Management (PALS) study was conducted with two cohorts, though both participated in peer support interventions (IRB# Pro00080875). A detailed description of the PALS protocol for this study has been published elsewhere [21]. PALS was approved by the IRB at the Medical University of South Carolina (MUSC). Study participants in the study provided written informed consent prior to engaging in study activities.

### 2.1. Study Participants and Recruitment

The PALS study largely enrolled SLE patients seen within the MUSC health system. However, individuals from the broader patient population in South Carolina were also included through referrals and advertisements placed throughout the community. All SLE patients included in PALS were African American women, over the age of 18, and could communicate in English. Excluded from the study were subjects with cognitive impairments, substance user disorders, and comorbid conditions that could impact their ability to communicate effectively (e.g., blindness or deafness).

### 2.2. Outcome Measures

Subjects in PALS completed questionnaires prior to taking part in any intervention activities. Outcomes of interest for this study included depression—measured by the Patient Health Questionnaire-8 (PHQ-8) [22], anxiety—measured by the Generalized Anxiety Disorder-7 (GAD-7) [23], and medical travel burden—assessed by a purpose-built survey.

### 2.3. Statistical Methods

Descriptive data for outcomes measure are presented. We ran unadjusted and adjusted linear regression models using the total PHQ-8 score or total GAD-7 score as the outcome variables. The adjusted model accounted for question 3 of the Systemic Lupus Activity Questionnaire (SLAQ). The SLAQ is a self-reported disease activity measure for SLE patients. Question 3 of the measure asks subjects to rate their disease activity in the previous 3-months on a 0—10 scale, where zero represents no activity and ten indicates highly active disease symptomology [24]. Statistical significance was determined when *p* < 0.05.

## 3. Results

The current analysis included baseline data from 100 SLE patients who participated in the PALS study (see Table 1). All study participants were African American women with a median age category of 25–34 years. The majority of study participants had a college degree (38.6%) and health insurance (92.1%). A little less than half were unemployed (46.1%). Baseline assessments for depression revealed that PALS participants reported a mean score on the PHQ-8 of 8.62 (±5.79), a baseline anxiety of 7.96 (±6.33) and an average self-reported disease activity measure by SLAQ of 5.30 (±2.74).

An assessment of healthcare-related travel burden at baseline revealed that only 9% of all study participants found it difficult or very difficult to travel to primary care visits. Perceived difficulty in traveling to rheumatology appointment was somewhat higher, with 13.7% of participants reporting that it was either difficult or very difficult to travel for those visits. Moreover, participants reported a mean travel distance of nearly 65.5 (±60.7) miles to their rheumatologist and that that travel took more than 70.3 (±64) minutes. Most participants (57.6%) reported that they traveled to between 1 and 2 healthcare-related appointments per week, and just over half (51.2%) indicated that their health insurance covers medical transportation. Similarly, 47.1% reported that it was either easy or very easy to arrange that transportation. Conversely, subjects reported missing on average about 2 (±6.57) appointments in the last year due to travel issues (see Table 2).

Linear regression models were used to assess the relationship between travel burden and depression and anxiety. Table 3 details the unadjusted model for the relationship between travel burden and depression. Frequency of healthcare visits was significantly associated with an increased PHQ-8 score (β = 1.6, *p* = 0.005). After adjusting for self-reported disease activity (SLAQ question 3), the relationship between travel burden and an additional healthcare visit remained statistically significant (β = 1.4, *p* = 0.01) and having to take time off for doctors’ appointments became statistically significant (β = 3, *p* = 0.03). After adjusting for disease activity, age, employment, income, and insurance, frequency of healthcare visits (β = 1.3, *p* = 0.02) and having to take time off for doctors’ appointments (β = 5.3, *p* = 0.03) remained significant (see Table 3).

Table 4 displays the unadjusted model when examining the effect of anxiety on travel burden. There was no statistically significant relationship between anxiety and travel in the unadjusted model. However, when controlling for disease activity, a statistically significant relationship was found between anxiety and requiring time off from work for healthcare appointments (β = 3.1, *p* = 0.03). In the model adjusting for disease activity, age, employment, and income, perceived difficulty in arranging transportation to healthcare appointments was found to have a statistically significant relationship with anxiety (β = 3.1, *p* = 0.04) (see Table 4). In addition, requiring time off from work remained significant (β = 4, *p* = 0.02).

## 4. Discussion

This study provides several key findings. First, travel burden had a significant effect on SLE patients’ self-reported depression. Second, requiring time off work for medical appointments showed a significant increase in SLE patients’ anxiety. Third, perceived difficulty in arranging medical transportation had an inverse statistically significant relationship with anxiety. Overall, the findings of this study seem to indicate that travel burden has a significant effect on depression and anxiety among African American women living with SLE.

These findings are consistent with a systematic review by Zhang et al. [25] that suggests that the prevalence of depression and anxiety is high in adult SLE patients. In addition, a meta-analysis by Moustafa et al. [26] found that depression and anxiety among SLE patients reached 78.6% and 71.6%, respectively. The results of the current study found that frequency of healthcare visits was significantly associated with an increased depression score. This is also consistent with literature that shows the relationship between depression and increased healthcare utilization in the United States. Of note, an estimated 85% of those living with depression are also living with one or more chronic conditions, and approximately 30% have four or more chronic conditions. Taken together, the resulting healthcare utilization results in those with depression is more than two times the healthcare costs of those who have not been diagnosed with depression [27]. The literature also recommends that rheumatologists screen for depression among SLE patients and make appropriate referrals to mental health services [25]. Though there was no statistically significant relationship between travel burden and anxiety, there was a statistically significant relationship between anxiety and requiring time off from work for healthcare appointments and perceived difficulty in arranging transportation to healthcare appointments.

In addition to the significant findings in this study, we recognize that there were a few limitations. First, this study used a small sample size (N = 100). Though this has been considered a limitation, the prevalence of SLE is not as high as other, more common chronic conditions, and the study sample reflects this distinction. Second, all study participants were African American women that mostly reside in a concentrated area of the southeast United States. These two factors may limit generalizability of study findings. Future research should consider a larger study population using a more regionally diverse population to determine if findings remain consistent.

## 5. Conclusions

In summary, these results show that there is a relationship between travel burden and depression and anxiety in African American women with SLE. This is important because travel burden remains a significant barrier to care for many SLE patients. Not only does inadequate access to care lead to greater direct healthcare costs but it can also result in the development or worsening of comorbid conditions. Compounding chronic conditions further complicate the treatment process for these patients and impede appropriate resource delivery.

## Figures and Tables

**Table 1 healthcare-09-01507-t001:** Baseline Demographics (N = 130).

n	100
Age (%)	
18–25	9 (10.1)
25–34	28 (31.5)
35–44	27 (30.3)
45–54	13 (14.6)
55–64	10 (11.2)
>65	2 (2.2)
Education (%)	
Less than high school	8 (9.1)
High School Grad	15 (17.0)
Some College	31 (35.2)
College Grad	34 (38.6)
Income (%)	
<$15 K	28 (31.5)
$15–$34.9 K	18 (20.2)
$35–$64.9 K	16 (18.0)
> or = $65 K	9 (10.1)
Other/don’t want to respond	18 (20.2)
Not married (%)	73 (82.0)
Unemployed (%)	41 (46.1)
Insured (%)	82 (92.1)

**Table 2 healthcare-09-01507-t002:** Access to Healthcare and Travel Burden Characteristics.

n	100
Difficulty Traveling to Primary Care Provider (%)	
Very Difficult	0 (0.0)
Difficult	8 (9.0)
Neither	20 (22.5)
Easy	24 (27.0)
Very Easy	37 (41.6)
Difficulty Traveling to Rheumatologist (%)	
Very Difficult	2 (2.3)
Difficult	10 (11.4)
Neither	22 (25.0)
Easy	22 (25.0)
Very Easy	32 (36.4)
Travel Issues Make Keeping Appointments Difficult (%)	
Very Difficult	3 (3.4)
Difficult	13 (14.9)
Neither	24 (27.6)
Easy	25 (28.7)
Very Easy	22 (25.3)
Frequency of Health Care Visits in a Week (%)	
0	25 (29.4)
1	34 (40.0)
2	15 (17.6)
3	7 (8.2)
4	3 (3.5)
7	1 (1.2)
Insurance Covers Medical Transportation (%)	43 (51.2)
Difficulty Scheduling Medical Transportation (%)	
Very Difficult	2 (3.8)
Difficult	2 (3.8)
Neither	24 (45.3)
Easy	13 (24.5)
Very Easy	12 (22.6)
Have to Take Time Off for Doctors’ Appointments (%)	50 (56.2)
Travel Affects Keeping Appointments (%)	0.30 (0.46)
Distance (in miles) to Rheumatologist (mean (SD))	65.47 (60.69)
Commute Time (in minutes) to Lupus Care (mean (SD))	70.33 (64.00)
Number of Appointments Missed as a Result of Transportation Issues (mean (SD))	2.26 (6.57)
Depression (PHQ-8) (mean (SD))	8.62 (5.79)
Anxiety (GAD-8) (mean (SD))	7.96 (6.33)
SLAQ (mean (SD))	4.56 (2.47)

**Table 3 healthcare-09-01507-t003:** Linear Regression Model of the Effect of Travel Burden Indicators on Depression (PHQ8).

	Model 1 ^a^		Model 2 ^b^		Model 3 ^c^	
	β (95%CI)	*p*-Value	β (95%CI)	*p*-Value	β (95%CI)	*p*-Value
Difficulty Traveling to Primary Care Provider	0.1 (−2.4, 2.6)	0.94	−0.1 (−2.8, 2.5)	0.9256	0.6 (−2.1, 3.3)	0.65
Difficulty Traveling to Rheumatologist	−1.6 (−4, 0.8)	0.19	−0.9 (−3.4, 1.6)	0.4565	−1.7 (−4.2, 0.9)	0.20
Distance (in miles) to Rheumatologist	−0.04 (−0.1, 0.02)	0.16	−0.03 (−0.1, 0.03)	0.2742	−0.02 (−0.1, 0.04)	0.45
Commute Time (in minutes) to Lupus Care	0.03 (−0.02, 0.1)	0.20	0.03 (−0.02, 0.1)	0.2564	0.02 (−0.03, 0.1)	0.43
Travel Issues Make Keeping Appointments Difficult	−0.4 (−2.2, 1.5)	0.69	−0.6 (−2.5, 1.3)	0.5256	−0.3 (−2.3, 1.8)	0.78
Frequency of Health Care Visits in a Week	1.6 (0.5, 2.6)	0.005	1.4 (0.3, 2.4)	0.0113	1.3 (0.2, 2.5)	0.02
Insurance Covers Medical Transportation	−1.2 (−3.9, 1.5)	0.38	−1.4 (−4.2, 1.3)	0.3038	−1.9 (−5, 1.2)	0.23
Have to Take Time Off for Doctors’ Appointments	2.6 (−0.2, 5.4)	0.067	3 (0.3, 5.7)	0.0321	5.3 (1.8, 8.8)	0.003
Travel Affects Keeping Appointments	0.4 (−3.2, 4)	0.81	1 (−2.7, 4.7)	0.586	0.8 (−2.9, 4.5)	0.66
Number of Appointments (in one year) Missed as a Result of Transportation Issues	0.1 (−0.2, 0.3)	0.65	0.02 (−0.2, 0.3)	0.8541	0.1 (−0.2, 0.3)	0.66
Transportation Issues Increase Stress	−0.6 (−3.5, 2.3)	0.68	−0.4 (−3.2, 2.4)	0.7618	−0.7 (−3.5, 2.1)	0.63
SLAQ			0.7 (0.1, 1.2)	0.0144	0.7 (0.1, 1.2)	0.014
Age					−0.3 (−1.4, 0.8)	0.63
Employment					−3.5 (−7.2, 0.2)	0.06
Income					0.3 (−0.6, 1.2)	0.48
Insurance					−3.8 (−10.3, 2.76)	0.25

^a^ Unadjusted; ^b^ Adjusting for disease activity using SLAQ.; ^c^ Adjusting for disease activity using SLAQ, Age, Employment, and Insurance.

**Table 4 healthcare-09-01507-t004:** Linear Regression Model of the Effect of Travel Burden Indicators on Anxiety (GAD8).

	Model 1 ^a^		Model 2 ^b^		Model 3 ^c^	
	β (95%CI)	*p*-Value	β (95%CI)	*p*-Value	β (95%CI)	*p*-Value
**Difficulty Traveling to Primary Care Provider**	1.14 (−1.6, 3.9)	0.41	2.2 (−0.8, 5.1)	0.14	3.1 (0.1, 6.1)	0.04
**Difficulty Traveling to Rheumatologist**	−1.7 (−4.3, 1)	0.21	−1.8 (−4.5, 0.9)	0.18	−2.54 (−5.2, 0.2)	0.07
**Distance (in miles) to Rheumatologist**	−0.03 (−0.1, 0.02)	0.25	−0.02 (−0.1, 0.04)	0.60	−0.002 (−0.1, 0.06)	0.96
**Commute Time (in minutes) to Lupus Care**	0.01 (−0.05, 0.1)	0.72	−0.003 (−0.06, 0.1)	0.92	−0.02 (−0.07, 0.04)	0.51
**Travel Issues Make Keeping Appointments Difficult**	0.3 (−1.6, 2.2)	0.76	−0.3 (−2.3, 1.6)	0.74	0.3 (−1.7, 2.3)	0.76
**Frequency of Health Care Visits in a Week**	0.7 (−0.4, 1.7)	0.21	0.6 (−0.4, 1.7)	0.23	0.5 (−0.6, 1.6)	0.34
**Insurance Covers Medical Transportation**	3 (−0.7, 6.8)	0.11	2.9 (−0.7, 6.6)	0.11	0.2 (−4.2, 4.6)	0.92
**Difficulty Scheduling Medical Transportation**	−0.3 (−1.9, 1.2)	0.68	−0.1 (−1.6, 1.4)	0.93	0.1 (−1.4, 1.5)	0.95
**Have to Take Time Off for Doctors’ Appointments**	2.4 (−0.5, 5.3)	0.10	3.1 (0.2, 5.9)	0.03	4 (0.5, 7.5)	0.02
**SLAQ**			0.52 (−0.1, 1.1)	0.10	0.04 (−0.02, 0.1)	0.20
**Age**					0.1 (−1, 1.2)	0.90
**Employment**					−2.7 (−6.8, 1.3)	0.18
**Income**					−0.9 (−1.8, −0.02)	0.05

^a^ Unadjusted; ^b^ Adjusting for disease activity using SLAQ.; ^c^ Adjusting for disease activity using SLAQ, Age, Employment, and Insurance.

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
