# Peer review of "The Effect of Travel Burden on Depression and Anxiety in African American Women Living with Systemic Lupus"

_healthcare, 2021, doi:10.3390/healthcare9111507_

Round 1

Reviewer 1 Report

The article is very interesting for publication. But it is necessary to modify some aspects.
-Add in the introduction the importance of podiatric problems in the feet, for example the following DOI reference: 10.1136 / bmjopen-2020-042627
-Modify tables 1 and 2 for a better understanding.
-Expand the discussion, adding the opinion of other authors and recent bibliography.

Reviewer 2 Report

The study highlights the importance of easy accessibility to the specialist physician when patients are affected by chronic diseases, especially systemic diseases with an unpredictable course.

I am not sure that the results are exclusively specific for Afro-American SLE patients. A control group of patients affected by other chronic diseases would have been interesting to compare.

Many other factors than disease activity may have contributed to the outcomes of the investigation. The statistical analyses should include other adjustments of importance, such as age, education level, marital and employment status. Are anxiety and depression perhaps more prevalent in the general population when you are older, unemployed and unmarried?

SLAQ is a self-reported disease activity measure for SLE. It is the only variable used to adjust the crude data of the linear regression models. A self-reported item may be inappropriate to use to investigate patients with anxiety and depression. Other disease activity measures should be included, e.g. SLEDAI.

Reviewer 3 Report

Thank you for the opportunity to review this interesting work.
Depression and anxiety play a big role in quality of life, especially in those with SLE. In addition to a greater burden of disease, patients of African descent in the United States have other social, financial and cultural burdens.
The authors recognize a connection between the frequency of check-ups and depression as well as difficulties in transportation, absenteeism due to doctor's appointments and anxiety.
Patients from 2 cohorts, CALLS (n = 30 inpatients) and PALS (n = 100 outpatients and people of African descent recruited for the study) were included. The description leaves open whether other ethnic groups are represented in the CALLS cohort. That should be made clearer. Tables 1 and 2 offer enough space to present the different cohorts in 2 separate columns. It is not clear to me why the inpatients report lower disease activity in the SLAQ than the PALS cohort. A medical activity assessment could further improve the data situation (e.g. SLEDAI). The high rate of unemployed test subjects amazes me. In addition, the effect of the downtime should not really be significant here. Is there an evaluation of the employed subjects only? Is there an evaluation of the income situation as a confounder for the measured values? The high frequency of health care visits in a week amazes me. In addition, 0 / week is certainly not correct, here further breakdowns such as every 2 weeks, monthly, in the quarter would be necessary. There is an association between health care usage frequency and depression. Couldn't it be that depressed patients go to the doctor more often because of the psychological stress (and not the other way around?). Data could help as to whether the rate of doctor contacts with depression is generally increasing in the USA. The negative correlation between fear and difficulty in organizing the trip is incomprehensible. I would expect a different result here. Has the question encryption been checked? Authors should better identify their position and employment, especially Authors 4, 5, 6, and 8. The * as the correspondent author is not found among the authors. Author 1 is incorrectly listed as 1.1. Individual indentations are incorrectly copied. I would be delighted to see this important work published in this journal, for example due to the proposed changes being a bit more understandable for outsiders.  
